# Contamination of Hotel Water Distribution Systems by *Legionella* Species: Environmental Surveillance in Campania Region, South Italy

**DOI:** 10.3390/microorganisms11071840

**Published:** 2023-07-20

**Authors:** Valeria Di Onofrio, Mariangela Pagano, Marco Santulli, Annamaria Rossi, Renato Liguori, Mirella Di Dio, Giorgio Liguori

**Affiliations:** 1International PhD Programme/UNESCO Chair “Environment, Resources and Sustainable Development”, Department of Sciences and Technologies, University of Naples “Parthenope”, Business District, Block C4, 80143 Naples, Italy; valeria.dionofrio@uniparthenope.it (V.D.O.); renato.liguori88@gmail.com (R.L.); 2ARPA Campania—Salerno Department—Via Lanzalone, 54/56, 84100 Salerno, Italy; m.pagano@arpacampania.it (M.P.); am.rossi@arpacampania.it (A.R.); 3School of Medicine and Health Sciences, “G. d’Annunzio” University of Chieti-Pescara, Via dei Vestini 31, 66100 Chieti, Italy; marco.santulli1@gmail.com; 4Department of Movement Sciences and Wellbeing, University of Naples “Parthenope”, Via Medina n. 40, 80133 Naples, Italy; giorgio.liguori@uniparthenope.it

**Keywords:** *Legionella* spp., environmental surveillance, hotel, water contamination

## Abstract

The COVID-19 pandemic period was marked by the absence or reduced circulation of some infectious diseases. Legionellosis may have been affected by the prevention measures adopted to counter COVID-19. Legionellosis is a form of pneumonia interstitial that is normally transmitted via aerosol-containing bacteria (genus *Legionella*), that could be present in contaminated water sources and is often associated with travel and with staying in hotels. In this work, the data of the environmental surveys carried out by ARPA Campania in accommodation facilities since 2019 were analyzed for a better understanding of the dispersion patterns of *L. pneumophila* associated with the environment and to evaluate the variation of the data during the pandemic period. The aim was to provide a better understanding of *Legionella* at different geographic scales and to define a predictive epidemiological method. Results: In 2019, the *Legionella* genus contaminated 37.7% of all tourist facilities evaluated. In 2020, the *Legionella* genus contaminated 44.4% of all tourist facilities evaluated. In 2021, the *Legionella* genus contaminated 54.2% of all tourist facilities evaluated. Conclusions: *Legionella pneumophila* was the most prevalent species in our community, serogroup 1 was the most frequently isolated and the most implicated risk factor of contamination was the temperature of water in circulation.

## 1. Introduction

The year 2020 was marked by the COVID-19 pandemic, which, during the year, caused 2,107,166 cases of infection and 74,159 deaths in Italy [1], and at the same time the absence or reduced circulation of some infectious diseases. Measles cases were not observed from April to December 2020 [2] and the flu disappeared throughout the 2020–2021 season [3]. The adopted prevention measures to counter COVID-19 have shown, therefore, to also be effective for other infectious pathogens and legionellosis may have been affected by these measures too. Legionellosis (Legionnaires’ disease), in fact, is a form of pneumonia interstitial which is normally transmitted via aerosol-containing bacteria, belonging to the genus *Legionella* that may be present in contaminated water sources, such as domestic hot water, fountains, equipment for respiratory therapy, cooling towers and other devices [4]. Legionellosis is often associated with travel and with staying in hotels [5,6].

Etiological agent of legionellosis is gram-negative and aerobic bacillus, named *Legionella* [7]. There are about 60 known *Legionella* species isolated from aqueous environments, although new species continue to be described [8]. Any *Legionella* species are able to cause legionellosis pneumonia but *L. pneumophila* is, by far, the most frequently associated with this disease [9]. This species comprises different serogroups, serogroup 1 being the major pathogen for humans, causing around 70–90% of infections [10]. *Legionella* species are found worldwide in natural (lakes, rivers, and thermal springs) and artificial water systems (city water pipes, water systems of buildings, hospitals, tourist facilities, and spas). These bacteria are present at the best concentrations in biofilms within plight systems and openings of water outlets; biofilms constitute a protective niche against water treatment procedures and stressful conditions. Stagnating warm water, water temperature between 20 and 45 °C, the presence of encrustations within the pipes, and corrosion provide a perfect habitat for the large growth of this bacterium [11].

In recent years, the increase in epidemics related to water consumption in industrialized countries, for example, is due to devices that produce aerosols, spas, etc., and has led to greater attention to the problems linked to the presence of pathogenic microorganisms in the treated water resources [12,13,14]. In Italy, according to the National Surveillance System for Legionnaires’ disease, the number of cases of legionellosis has been progressively increasing, and the same trend was registered worldwide [15]. In the last thirty years, the number of legionellosis cases in Italy has increased steadily, although it is a largely under-reported disease as surveillance systems are not always used adequately. However, in conjunction with the pandemic, 2020 had recorded the lowest incidence since 2017 [16], the year during which 2014 cases of legionellosis were notified (of which 1981 confirmed and 33 probable), equal to 33.2 cases/million inhabitants [17]. The notification of cases of legionellosis to public health authorities in Italy has been mandatory since 1983. The identification of an isolated case of legionellosis triggers investigation procedures with confirmation stages, notification, and epidemiological study geared towards detecting the origin of the exposure to the pathogen. Similarly, two or more cases with presumed common origin within a two-year period prompt a series of verification steps aimed at determining the common source of the infection [18]. In this context, and based on the hypotheses emerging from descriptive analysis and a targeted anamnesis, environmental surveys on water/air networks and suspicious equipment are carried out. Based on the results of microbiological monitoring, remedial actions on contaminated water plumbing and/or aeraulic pipe are planned, following two main types of criteria: the concentration of *Legionella* and the percentage of positive samples [18]. The National Register of Legionellosis was established and managed by the “Istituto Superiore di Sanità” (National Public Health Institute) on the basis of surveillance forms submitted by physicians to the Local Health Units for each Legionnaires’ disease case. Since 2005, a network of reference regional centers responsible for microbiological diagnoses and environmental control of legionellosis within their respective regional areas has been established [18,19]. ARPA Campania (“Agenzia Regionale per la Protezione dell’Ambiente della Campania”) is the reference regional center for the Campania region, in Southern Italy. ARPA Campania is accredited in accordance with ISO/IEC 17025:2005 and ISO 11731:2017 standards for *Legionella* testing [20,21] and carries out its activities following confirmed clinical cases and/or control actions prescribed by the Health Regional Authorities, with the purpose of identifying and preventing possible outbreaks. ARPA-Campania follows the ECDC (European Centre for Disease Prevention and Control) Legionnaires’ disease surveillance network called European Legionnaires’ Disease Surveillance Network (ELDSNet) for the investigation of travel-associated epidemic clusters as well as hospital-acquired to highlight the risk factors and interrupt the transmission chain [22,23]. In this work, the data of the environmental surveys carried out in the Campania region by ARPA Campania in accommodation facilities since 2019 were analyzed for a better understanding of the dispersion patterns of *L. pneumophila* associated with the environment and to evaluate the variation of the data during the COVID-19 pandemic period. The aim was to provide a better understanding of this opportunistic pathogen at different geographic scales and to define a method that provides possible scenarios of the spread of the disease and which can be adopted by the public health authorities of the region in the coming years.

## 2. Materials and Methods

### 2.1. Study Area, Type of Structures Analyzed and Samples Taken

The collection of water samples was conducted from 2019 to 2021 in all provinces of the Campania region, Southern Italy (Naples, Caserta, Avellino, Salerno, and Benevento). A total of 95 accommodation facilities were monitored. Samples were taken from both cold and warm water systems. The mean hot water temperature was 38 °C and the mean cold water temperature was 22 °C. All water samples do not come from tanks but come from mains water. A total of 1048 samples were analyzed. The water supplied to the accommodation facilities (which contains free chlorine as a residual disinfectant of drinking water) comes from the public network of the cities in which they are located. The accommodation facilities examined were: hotels, tourist villages, campsites, guest houses, bed and breakfasts, and spas (in the latter, no samples of thermal water were analyzed; thermal water means the water that comes from the source and flows directly from the subsoil).

### 2.2. Sample Collection

According to the Italian national guidelines, the laboratory staff collected 1 L of water for each sample, which was placed inside sterile polyethylene bottles enriched with 0.01% sodium thiosulphate to neutralize the action of chlorine. Samples were collected by simply simulating the shared use of water by facility users. Therefore, the filters from the taps have not been removed and neither flaming nor flushing have been carried out. Once collected, the samples were uniquely identified and annotated on a spreadsheet, after which they were transported at a suitable temperature and protected from light, taking care to separate the hot water samples from those of cold water, and were sent to the ARPA Campania laboratories accredited according to the UNI CEI EN ISO/IEC 17025 standard [20]. No samples were damaged during transport, therefore, all samples were included in the study and analyzed.

### 2.3. Microbiological Analysis and Identification

The ARPA Campania Laboratory carries out its activities for the research environmental sources of contagion for *Legionella* spp. after the reporting of individual cases or clusters of cases who have stayed at the same site by the National Public Health Institute. The laboratory is accredited according to the UNI CEI EN ISO/IEC 17025 standard [20] for the isolation, quantification, and typing of *Legionella* spp. in environmental samples with the cultural method, according to the UNI EN ISO 11731:2017 method [21], and for sampling according to the UNI EN ISO 19458:2006 standard (Figure 1).

In line with the National Public Health Institute recommendations, the ARPA Campania Laboratory is equipped with a real-time PCR system for rapid routine analysis of environmental samples related to epidemic outbreaks, for which the timeliness of investigations is even more necessary in order to implement the appropriate control measures for the containment of cases of the disease.

### 2.4. Data Analysis

The collected data from the study were imported into a Microsoft Excel 2016 file. Data were curated by analyzing the metadata of the samples (hotel, sample code, date, installation, etc.) to detect and eliminate duplicates and inconsistencies, and, finally, were considered values for 1048 samples. We used a two-tailed chi-squared test for the qualitative data analysis (*Legionella* presence/absence, temperature) and the *t*-test for quantitative data (bacterial counts). Results were considered statistically significant at *p* values < 0.05. All statistical analyses were also performed in Microsoft Excel 2016.

## 3. Results

### Colonization by Legionella: Levels of Contamination and Species Distribution

In this work, the last three years of monitoring for *Legionella* were considered, before and during the COVID-19 pandemic.

Every year and in all structures, in addition to the hot water circuits, the samples taken from the cold water circuits were also analyzed. Always, in the cold water samples, the presence of *Legionella* spp. was clearly lower than those of hot water samples and with charges always lower than 5000 CFU/L.

In 2019, n. 53 accommodation facilities were monitored, differently distributed throughout the regions: Naples (n. 37), Salerno (n. 11), Avellino (n. 2), Caserta (n. 2), and Benevento (n. 1). *Legionella* genus contaminated 20 (37.7%) of all tourist facilities evaluated. A total of 418 water samples, taken from different hot and cold water systems of hotels, were analyzed. Out of the 418 samples, 118 (28.2%) were positive for *L. pneumophila*. Of the 118 positive samples, 98 samples came from hot water circuits, and the remainder from cold water circuits. Serological typing of the 118 *L. pneumophila* isolate revealed that 55 (46.7%) are *Legionella pneumophila* serogroup 1; 1 (0.8%) is *Legionella pneumophila* serogroup 3; 16 (13.6%) are *Legionella pneumophila* serogroup 6; 28 (23.7%) are *Legionella pneumophila* serogroup 8; 1 (0.8%) is *Legionella pneumophila* serogroup 9; 8 (6.8%) are *Legionella pneumophila* serogroup 10; and 9 (7.6%) are *Legionella pneumophila* serogroup 14 (Figure 2). 

Of the 418 samples tested, the 118 positive samples consisted of 6 (5.1%) samples containing a bacterial load between 10 and 10^3^ CFU/L, and 112 (94.9%) samples containing a bacterial load between 10^3^ and 10^4^ CFU/L (Table 1). 

Therefore, 112 (94.9%) counts exceeded 1000 CFU/L, the limit of public health, requiring legally mandated disinfection measures. The highest counts observed in 2019 were over 8.0 × 10^4^ CFU/L. *L. pneumophila* was isolated from 118 of 418 water samples derived from 21/53 hotels (39.6%) (Table 2).

In 2020, n. 18 accommodation facilities were monitored, differently distributed throughout the region: Naples (n. 12), Salerno (n. 4), Avellino (n. 0), Caserta (n. 0), and Benevento (n. 2). *Legionella* genus contaminated 8 (44.4%) of all tourist facilities evaluated. A total of 273 water samples taken from different hot and cold water systems of hotels were analyzed. Out of the 273 samples, 62 (22.7%) were positive for *L. pneumophila*. Of the 62 positive samples, 48 samples came from hot water circuits, and the remainder from cold water circuits. Serological typing of the 62 *L. pneumophila* isolate revealed that 53 (85.5%) are *Legionella pneumophila* serogroup 1; 4 (6.5%) are *Legionella pneumophila* serogroup 3; and 5 (8.0%) are *Legionella pneumophila* serogroup 8 (Figure 3). 

Among the 62 positive samples, 7 (11.3%) samples contained a bacterial load between 10 and 10^3^ CFU/L, and 55 (88.7%) samples contained a bacterial load between 10^3^ and 10^4^ CFU/L (Table 1). Therefore, 55 (88.7%) counts exceeded 1000 CFU/L, the limit of public health, requiring legally mandated disinfection measures. The highest counts observed in 2020 were over 7.8 × 10^4^ CFU/L. *L. pneumophila* was isolated from 62 of 273 water samples derived from 8/18 hotels (44.4%) (Table 2).

In 2021, n. 24 accommodation facilities were monitored, differently distributed throughout the region: Naples (n. 14), Salerno (n. 4), Avellino (n. 3), Caserta (n. 3), and Benevento (n. 0). *Legionella* genus contaminated 13 (54.2%) of all tourist facilities evaluated. A total of 357 water samples taken from different hot and cold water systems of hotels were analyzed. Out of the 357 samples, 89 (24.9%) were positive for *L. pneumophila*. Of the 89 positive samples, 75 samples came from hot water circuits, and the remainder from cold water circuits. Serological typing of the 89 *L. pneumophila* isolate revealed that 50 (56.2%) are *Legionella pneumophila* serogroup 1; 4 (4.5%) are *Legionella pneumophila* serogroup 6; 13 (14.6%) are *Legionella pneumophila* serogroup 8; 1 (1.1%) is *Legionella pneumophila* serogroup 10; 7 (7.9%) are *Legionella pneumophila* serogroup 11; 10 (11.2%) are *Legionella pneumophila* serogroup 12; and 4 (4.5%) are *Legionella* spp. (Figure 4). 

Among the 89 positive samples, 9 (10.1%) samples contained a bacterial load between 10 and 10^3^ CFU/L, and 80 (89.9%) samples contained a bacterial load between 10^3^ and 10^4^ CFU/L (Table 1). Therefore, 80 (89.9%) counts exceeded 1000 CFU/L, the limit of public health, requiring legally mandated disinfection measures. The highest counts observed in 2021 were over 8.0 × 10^4^ CFU/L. *L. pneumophila* was isolated from 89 of 357 water samples derived from 13/24 hotels (54.2%) (Table 2).

## 4. Discussion

The annual epidemiological reports of the European Centre for Disease Prevention and Control confirmed an important increase in Legionellosis cases in the last years [24]. Notification rates remained heterogeneous across the EU/EEA (European Union/European Economic Agreement), varying from fewer than 0.5 cases per 100,000 population to 5.7 cases per 100,000 population, with the highest rate reported by Slovenia. Four countries (France, Germany, Italy, and Spain) accounted for 72% of all notified cases. Males aged 65 years and older were most affected (7.1 cases per 100,000 population). The number of reported cases to the travel-associated surveillance scheme decreased by 67% in 2020 compared with 2019. Only 10% of cases were culture-confirmed (10%), likely leading to an underestimation of diseases caused by *Legionella* species other than *Legionella pneumophila* [25]. Members of the *Legionella* genus contaminated 45.4% of all tourist facilities evaluated in this study. This contamination rate is lower than reported in studies conducted in Hungary (72%) [26], Italy (66.9%) [27], Greece (75%) [28], and the Netherlands (85%) [29] but remarkably higher than rates reported in studies performed in Croatia (27.2%) [30]. Several non-exclusive factors might explain this apparent discrepancy between studies and countries, such as the number of facilities or samples analyzed in each study, the covered period, and the evaluated installation type. In our study, we included a number of facilities and samples, over a 3-year period, which coincided with the COVID-19 pre-pandemic and pandemic period. We selected touristic facilities based on previous Legionnaires’ disease episodes, thereby introducing a bias. Our results may be underestimated due to the smaller number of cases reported during the pandemic period and the consequent lower number of controlled facilities. *Legionella pneumophila* was the most prevalent species in our community. As regards the individual serogroups, the highest percentage of positivity was found for *L. pneumophila* serogroup 1. This result seems to be in line with what was found in the clinical diagnosis, according to which *L. pneumophila* serogroup 1 was the most frequently isolated and the species associated with most human cases in Europe and North America [31]. Our investigation shows that the most implicated risk factor of contamination was the temperature of water in circulation. Mean contamination levels were highest in the hot water systems of hotels. The high temperature was associated both with the level of contamination and the percentage of positive samples. Regarding the collection and processing of data related to the year 2022, the data of samples collected in the accommodation facilities of Campania are currently being processed. But it is already quite clear that the end of the pandemic period has influenced the spread and contagion of legionellosis. Furthermore, during the year 2022, hospitals have adopted a differential diagnostic protocol for cases of pneumonia which includes both the test for the search for SARS-CoV-2 (COVID-19) and the rapid test for the search for urine antigen of Legionella. The differential diagnosis has led to a significant surge in the total number of cases of legionellosis diagnosed in Campania, a phenomenon that has been monitored by the Regional Health Department together with ARPAC. 

Environmental surveillance of *Legionella*, entrusted to the Regional Reference Center for legionellosis incardinated in the Arpac’s provincial Department of Salerno, focuses on the sites frequented by the patients who have developed the disease. It consists in identifying the so-called “critical points” of the water and air conditioning systems (where it is easier for the bacterium, responsible for the infection, to nest and proliferate) from which to take and analyze samples of environmental matrices (water, air, biofilm, and sediments). The trend, in recent years, of legionellosis cases reports received by ARPA Campania, shows that from 116 in 2017 it has gone up to 135 in 2018, 124 in 2019, 60 in 2020, and 83 in 2021. The last two years have seen a sharp decrease in reports, especially those related to the tourism–accommodation world, due to the COVID-19 emergency. The activity of the Legionnaires’ disease laboratory is one of the most significant examples of collaboration between the Environmental Agency and the health system. In this case, the possibility that the institution in charge of environmental controls contributes to relieving the pressure on the health system is particularly evident; therefore, the environmental protection activities also have an impact on human health, in the short and long term. Since in the severe form there is a picture of interstitial pneumonia, the differential diagno-sis with SARS–CoV-2 must absolutely be made. Both the investigation activities, carried out to identify the environmental source of infection, and the subsequent surveillance ac-tivities, carried out to verify the effectiveness of the remediation interventions on plants contaminated with Legionella consist in the search for Legionella spp. in the environmental matrices (biofilm, encrustations, water, air) taken from the places frequented by the patient, at the critical points of the water and air conditioning systems. There is a European Surveillance System (the ELDSNet headed by the ECDC based in Stockholm) for the control of the legionellosis of travelers, which has the Istituto Superiore di Sanità as the coordinating center for Italy. When a tourist contracts the disease after staying in a receptive facility, the Stockholm Center reports the case of the disease to the ISS, which alerts the territorially competent ASL and the Regional Reference Laboratory for the research of the environmental source of contagion of the disease.

Our study provides a partial overview of the presence of *Legionella* in hotel water systems. It would be interesting, and equally important, to broaden the research by also addressing it to other types of structures, as well as evaluating if the trend of legionellosis over the years could somehow reverse itself. To make this happen, it may be useful, as previously mentioned, to implement control systems. Another point on which to pay attention could concern the awareness of those who manage and frequent the accommodation facilities on the effective existence of the risk of contamination of the water systems by *Legionella*. Legionnaires’ disease is a real public health problem that also seems to be underestimated, as surveillance systems are not used adequately in all countries. Furthermore, the underestimation of cases is also linked to an objective difficulty in culturing the bacterium [32]. However, the data available to us indicate the need to further investigate this area of research in order to ensure public health.

Furthermore, a last, but not least, point to focus on is the aspect related to tourism. The Campania region has natural and cultural resources recognized and appreciated all over the world, therefore it represents a pole of tourist attraction and as such it must adapt to the quality and safety standards existing in the most advanced European countries. This would allow it to reach the condition of “Legionella free” in the structures of the accommodation and community facilities of the Region. To achieve this goal, the collaboration of multidisciplinary professionals (biologists, health professionals, engineers, installers, etc.) will be necessary for the definition of ad hoc surveillance plans based on the peculiarities of the thermo-hydraulic systems. The construction of an integrated network of various professionals is essential, in fact, to approach the problem on several fronts. In fact, if on the one hand effective clinical sur-veillance is necessary for the early diagnosis of the disease, on the other hand careful en-vironmental surveillance is essential through the development and implementation of risk assessment plans and self-monitoring programs in the structures at risk of contamination. This is equivalent to guaranteeing correct design, installation and maintenance of the wa-ter and air conditioning systems of the accommodation facilities.

Below, we report, in fact, some examples of design and maintenance interventions to be implemented in water systems.

The following prevention measures for water systems are crucial.

Design the water system by keeping the hot water and cold water pipes separate;Keep the system active at temperatures that do not allow growth of *Legionella* (<20 °C–>60 °C);Avoid water stagnation;Keep the system clean;Carry out microbiological analyses for the search for *Legionella* in the critical points of the plants.

Some preventive measures are also necessary for air conditioning systems:Periodically inspect the system to check the status of the ducts;Periodically inspect the system to check the cleanliness and maintenance of humidifiers and evaporative towers;Clean the cooling towers at least twice a year;Change the filters at predetermined intervals, and completely and regularly clean all parts of the humidifier;Carry out periodic microbiological analyses for the research of *Legionella* species.

Furthermore, it would be useful to investigate a further aspect, namely the relationships between Legionella and the various elements of its habitat, such as the presence of iron, an indispensable element for the growth of Legionella, or copper, which appears to inhibit the growth of Legionella. 

## Figures and Tables

**Figure 1 microorganisms-11-01840-f001:**
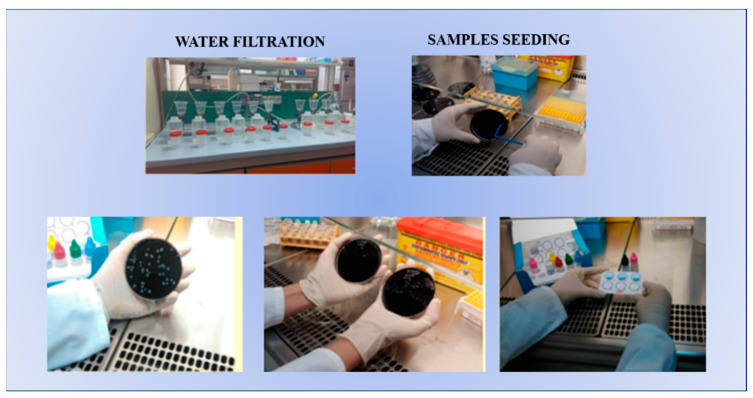
Sample analysis (UNI EN ISO 11731:2017: culture methods for the isolation of *Legionella* and estimation of their numbers in water samples).

**Figure 2 microorganisms-11-01840-f002:**
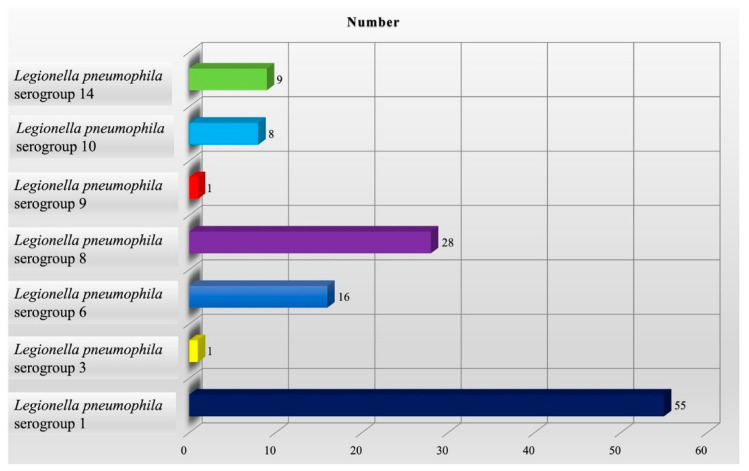
Distribution of *Legionella* species in 2019.

**Figure 3 microorganisms-11-01840-f003:**
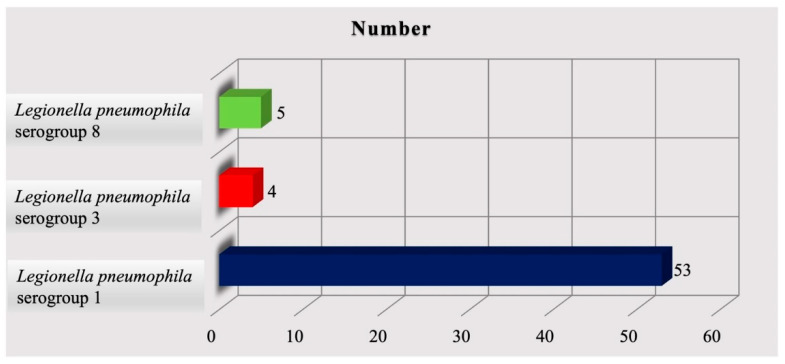
Distribution of *Legionella* species in 2020.

**Figure 4 microorganisms-11-01840-f004:**
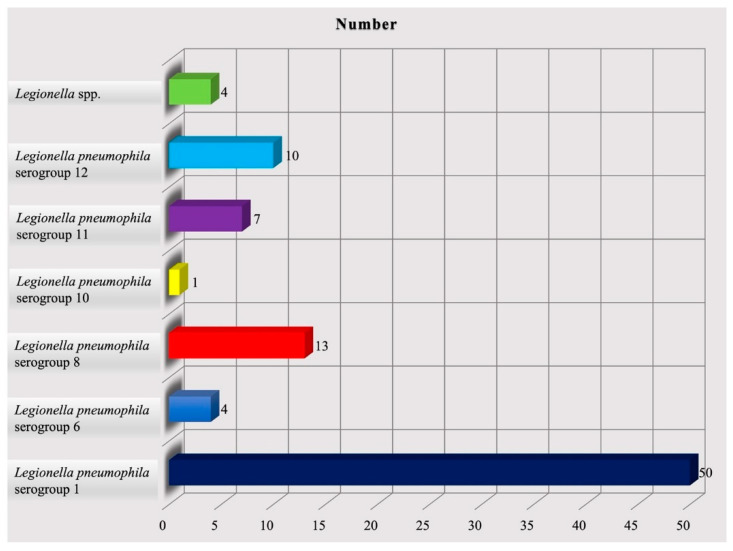
Distribution of *Legionella* species in 2021.

**Table 1 microorganisms-11-01840-t001:** Number of structures analyzed, number of positive samples and bacterial counts divided by year.

Year	Number of Structures Examined	Number of Samples Collected	Number of Positive Samples	Bacteria Count (CFU/L)
2019	53	418	118	In 112 samples, between 10^3^ and 10^4^ In 6 samples, between 10 and 10^3^
2020	18	273	62	In 55 samples, between 10^3^ and 10^4^ In 7 samples, between 10 and 10^3^
2021	24	357	89	In 80 samples, between 10^3^ and 10^4^ In 9 samples, between 10 and 10^3^

**Table 2 microorganisms-11-01840-t002:** Highest counts and number of hotels where positive samples were found divided by year.

Year	Highest Counts (CFU/L)	N° of Hotel with Positive Samples/n° of Hotel Examined
2019	Over 8.0 × 10^4^	21/53
2020	Over 7.8 × 10^4^	8/18
2021	Over 8.0 × 10^4^	13/24

## Data Availability

Data available on request due to restrictions.

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
