# Peer review of "Contamination of Hotel Water Distribution Systems by Legionella Species: Environmental Surveillance in Campania Region, South Italy"

_microorganisms, 2023, doi:10.3390/microorganisms11071840_

Round 1
Reviewer 1 Report
Dear Authors,
This is an interesting and well constructed paper. The Introduction is long. Despite it length it would be interesting to additional information on the increase in Legionella seen over the years. You note it has increased, but not by how much and give the numbers for 2017 but not previous years.. A sentence about how much higher this was than years past would be of interest. If possible would try to tighten up the Introduction without losing important background information relevant to your study. The last sentence of the introduction regarding the aim of the study could be improved and be made more clear for nonepidemiologists.
In paragraph 2.1 clarify what "thermal water" means (hot water circuits were analyzed). Why was "thermal water" excluded from the study?
I suggest clarifying that the 112 (94.9%) >1,000 cfu in 2019 was of the 118 + samples of a total of 418. This sentence is misleading if someone is skimming the article. A simple way to improve would be to connect this sentence to the sentence preceding Table 1 and have them in the same paragraph.
It would be interesting to the reader to understand how often Legionella was found only in the hot water circuit (and what the temperature was), in the cold water circuit and what the temperature was and if this differed compared to the hot and cold water circuits where no Legionella was found. You report more Legionella was found in the hot water circuits and it related to the temperature, but providing the data would improve the paper.
I appreciated the information on prevention measures. How do we know these work? Can you provide information that for hotels where these type of measures have been implemented, lower cfu of no Legionella was found, compared to hotels that have not implemented these measures?
Some of the wording is awkward, but the paper is easily understandable to a native English speaker.
Author Response
Response to Reviewer 1 Comments
Point 1: This is an interesting and well constructed paper. The Introduction is long. Despite it length it would be interesting to additional information on the increase in Legionella seen over the years. You note it has increased, but not by how much and give the numbers for 2017 but not previous years. A sentence about how much higher this was than years past would be of interest. If possible would try to tighten up the Introduction without losing important background information relevant to your study. The last sentence of the introduction regarding the aim of the study could be improved and be made more clear for nonepidemiologists.
Response 1: Thanks for the valuable suggestion. We have inserted a sentence that summarizes the situation inherent in the last 30 years concerning the number of legionella cases in Italy and, in particular, relates the data of 2020 with those of 2017, the year in which, before 2020, were the lower number of cases. We have also changed the last sentence of the introduction.
Point 2: In paragraph 2.1 clarify what "thermal water" means (hot water circuits were analyzed). Why was "thermal water" excluded from the study?
Response 2: The hot water circuits coming from the aqueduct were analysed, therefore, it is not thermal water. We did not analyze the thermal water samples as there were not sources of thermal water in all the structures examined. In any case, now we have included a definition of thermal water.
Point 3: I suggest clarifying that the 112 (94.9%) >1,000 cfu in 2019 was of the 118 + samples of a total of 418. This sentence is misleading if someone is skimming the article. A simple way to improve would be to connect this sentence to the sentence preceding Table 1 and have them in the same paragraph..
Response 3: We have slightly modified the sentence in question, to make the reading of the data clearer.
Point 4: It would be interesting to the reader to understand how often Legionella was found only in the hot water circuit (and what the temperature was), in the cold water circuit and what the temperature was and if this differed compared to the hot and cold water circuits where no Legionella was found. You report more Legionella was found in the hot water circuits and it related to the temperature, but providing the data would improve the paper.
Response 4: We proceeded to enter the number of positive samples from the two water circuits (hot/cold). In this way the results are certainly more complete. We have inserted the average temperatures at which the samples of hot and cold water were taken.
Point 5: I appreciated the information on prevention measures. How do we know these work? Can you provide information that for hotels where these type of measures have been implemented, lower cfu of no Legionella was found, compared to hotels that have not implemented these measures?
Response 5: Currently we have no data available, but the evaluation of this aspect is one of our future intentions.
Reviewer 2 Report
Contamination of hotel water distribution systems by Legionella species: environmental surveillance in Campania Region, South Italy
This paper reports a considerable amount of results about Legionella species in water distribution system, however the results could be better described and in the reviewer opinion the discussion was not very concise. Therefore, some modifications would be done before publication.
The results could be better described. Sometimes it is not clear what is the main information from the sentence (eg row 5-6 in the results section). As I suggest to better describe results maybe with some characteristic of the waste system or other water parameters.
I agree with authors that further research about relationship of Legionella species with different (page 9 row 19 ) characteristic of the water system would be useful.
The English writing should be further polished.
Author Response
Response to Reviewer 2 Comments
Point 1: This paper reports a considerable amount of results about Legionella species in water distribution system, however the results could be better described and in the reviewer opinion the discussion was not very concise. Therefore, some modifications would be done before publication.
Response 1: We've tweaked the results slightly and revised and improved the discussions with a few small changes.
Point 2: The results could be better described. Sometimes it is not clear what is the main information from the sentence (eg row 5-6 in the results section). As I suggest to better describe results maybe with some characteristic of the waste system or other water parameters.
Response 2: We have made the sentence you reported clearer.
We have specified that in all cases the waters from which the samples were taken come from the water mains and not from tanks. We have also provided additional information about the water characteristics, such as the water temperature (hot and cold).
Point 3: I agree with authors that further research about relationship of Legionella species with different (page 9 row 19 ) characteristic of the water system would be useful.
Response 3: We thank you, we are glad that your thoughts are in line with ours.
Round 2
Reviewer 1 Report
Dear Authors, I believe the changes made improve the article. The article could benefit from having a native English speaker review and revise. For the most part, I understood what was trying to be conveyed, but some wording is not fluid. A few specific suggestions follow:
Second sentence of introduction needs grammatical help. It does not stand alone. Could change to Measle cases were not observed from …. The intro particularly the first paragraph could still benefit from being made more concise. Not all of the information relayed is relevant to the paper, for example the first outbreak etc….
The sentence after ref 15 is poorly worded and I do not know what it is trying to convey outside of legionella being largely under-reported. Why it is underreported is not made clear.
Suggest changing supposed to presumed common origin in the sentence preceding ref 18
Many other wording changes could be helpful to native English speakers.
See comments to authors
Author Response
Response to Reviewer 1 Comments
Point 1: Second sentence of introduction needs grammatical help. It does not stand alone. Could change to Measle cases were not observed from ….
Response 1: Thank you for your suggestion. We have corrected what you have reported.
Point 2: The intro particularly the first paragraph could still benefit from being made more concise. Not all of the information relayed is relevant to the paper, for example the first outbreak etc….
Response 2: We've removed some unnecessary information and made the introduction more concise.
Point 3: The sentence after ref 15 is poorly worded and I do not know what it is trying to convey outside of legionella being largely under-reported. Why it is underreported is not made clear.
Response 3: We have clarified this point.
Point 4: Suggest changing supposed to presumed common origin in the sentence preceding ref 18.
Response 4: Thanks for your suggestion.
Point 5: Many other wording changes could be helpful to native English speakers.
Response 5: We've made some wording more fluid.